Manuscript prepared for Geosci. Model Dev.
with version 2015/04/24 7.83 Copernicus papers of the LaTeX class copernicus.cls.
Date: 7 April 2016

# Development of aroCACM/MPMPO 1.0: A Model to Simulate Secondary Organic Aerosol from Aromatic Precursors in Regional Models

Matthew L. Dawson[1], Jialu Xu[2,3], Robert J. Griffin[2], and Donald Dabdub[1]

[1]Department of Mechanical and Aerospace Engineering, University of California, Irvine, CA 92697, USA
[2]Department of Civil and Environmental Engineering, Rice University, Houston, TX 77005, USA
[3]ACS Engineering, 16225 Park Ten Place Suite 110, Houston, TX, 77084, USA

*Correspondence to:* Donald Dabudb (ddabdub@uci.edu)

**Abstract.** The atmospheric oxidation of aromatic compounds is an important source of secondary organic aerosol (SOA) in urban areas. The oxidation of aromatics depends strongly on the levels of nitrogen oxides ($NO_X$). However, details of the mechanisms by which oxidation occurs are only recently being elucidated. Xu et al. (2015) developed an updated version of the gas-phase Caltech
Atmospheric Chemistry Mechanism (CACM) designed to simulate toluene and *m*-xylene oxidation in chamber experiments over a range of $NO_X$ conditions. The output from such a mechanism can be used in thermodynamic predictions of gas-particle partitioning leading to SOA. The current work reports the development of a model for SOA formation that combines the gas-phase mechanism of Xu et al. (2015) with an updated lumped SOA partitioning scheme (MPMPO) that allows partitioning
to multiple aerosol phases and that is designed for use in larger scale three-dimensional models. The resulting model is termed aroCACM/MPMPO 1.0. The model is integrated into the University of California, Irvine – California Institute of Technology (UCI-CIT) airshed model, which simulates the South Coast Air Basin (SoCAB) of California. Simulations using 2012 emissions indicate that "low-$NO_X$" pathways to SOA formation from aromatic oxidation play an important role, even in
regions that typically exhibit high $NO_X$ concentrations.

## 1 Introduction

Atmospheric aerosol particles negatively affect human health, contribute to reduced visibility, and impact Earth's climate through their ability to scatter and absorb radiation and affect cloud properties (Finlayson-Pitts and Pitts, 2000; Seinfeld and Pandis, 2006). Secondary organic aerosol (SOA) ac-
20 counts for a significant fraction of atmospheric aerosol mass (Hallquist et al., 2009; Kanakidou et al., 2005; Kroll and Seinfeld, 2008). In urban areas, the gas-phase oxidation of aromatic hydrocarbons is thought to be an important contributor to SOA formation, although the fundamental mechanisms are only recently being elucidated (Im et al., 2014; Ng et al., 2007; Song et al., 2005; Xu et al., 2015).

In urban areas, aromatics are an important constituent of the mix of volatile organic compounds (VOCs) and are emitted primarily from industrial operations, fuel evaporation, and vehicle exhaust, with smaller contributions from biomass burning and other sources (Karl et al., 2009). In addition, there is evidence of biogenic sources of aromatic compounds that may be important in rural areas (Gratien et al., 2011; White et al., 2009).

The atmospheric oxidation of aromatics is thought to proceed primarily by reaction with hydroxyl radical (OH), but nitrate ($NO_3$) and chlorine (Cl) radicals readily oxidize some substituted aromatics as well (Finlayson-Pitts and Pitts, 2000). The oxidation pathway (addition to the ring as opposed to hydrogen atom extraction from a side-chain) depends on the oxidant and the extent of substitution. Once initiated, oxidation proceeds through a variety of pathways that depend strongly on the level of nitrogen oxides ($NO_X$) that are present. In most cases, SOA yields from aromatics are negatively correlated with $NO_X$ concentrations (Ng et al., 2007; Song et al., 2005; Xu et al., 2015), though exceptions have been reported (Im et al., 2014).

In order to simulate the formation of SOA and other secondary pollutants such as ozone, air quality models require chemical mechanisms that predict gas-phase oxidation chemistry over a wide range of $NO_X$ concentrations in a computationally efficient manner. Such mechanisms can be highly reduced ("lumped") or highly specific depending on computational demands and application. For example, the computationally efficient mechanism of Carter et al. (2012) uses non-specific reactions to simulate the formation of secondary products, but does not track oxidation products specifically. On the other hand, Im et al. (2014) developed the comprehensive UNIPAR model to predict SOA formation from toluene and 1,2,3-trimethylbenzene. UNIPAR uses the detailed Master Chemical Mechanism (MCM (Jenkin et al., 2003; Saunders et al., 2003)) to simulate aromatic SOA formation based on a lumped equilibrium-partitioning scheme that accounts for liquid-liquid phase separation, aqueous-phase oligomerization, and organo-sulfate formation. The UNIPAR output in conjunction with outdoor chamber experiments performed over a range of $NO_X$ conditions suggest an important role for aerosol-phase oligomerization in aromatic SOA formation.

As an intermediate approach to the non-specific and comprehensive methods mentioned above, Xu et al. (2015) updated the Caltech Atmospheric Chemical Mechanism (CACM) to include SOA formation from the gas-phase oxidation of toluene and *m*-xylene and evaluated the model through comparison to chamber experiments. When combined with a partitioning model based on absorption theory (Pankow, 1994), the updated CACM simulated SOA formation accurately over a range of $NO_X$ conditions, without the need for separate low- and high- $NO_X$ parameters. In addition, the reduced complexity of CACM compared to the fully explicit MCM decreases computational cost, making it ideal for incorporation into larger-scale models.

This work reports results from the development of an updated SOA-formation mechanism, aro-CACM/MPMPO 1.0, based on the initial work of Xu et al. (2015) and an updated version of the Model to Predict the Multi-phase Partitioning of Organics (MPMPO, (Griffin et al., 2003)). The

testbed for this mechanism is the University of California, Irvine – California Institute of Technology (UCI-CIT) Airshed model, which simulates the South Coast Air Basin (SoCAB) of California. This work is presented in two parts, each with the aim of improving SOA predictions in regional models covering a range of $NO_X$ conditions. First, the vapor pressure calculations for all lumped
SOA species in the model were updated using the SIMPOL.1 method of Pankow and Asher (2008), and lumped SOA species were redistributed based on thermodynamic properties and relative abundance. Second, the updated CACM of Xu et al. (2015) was incorporated along with appropriate lumped SOA species and a treatment of aqueous-phase partitioning of aromatic-derived SOA.

## 2  Model Description

### 2.1  UCI-CIT Airshed Model

The UCI-CIT model is an Eulerian chemical transport model that solves simultaneously the advection/diffusion equation coupled with emission, deposition, and chemistry terms. The model domain is divided horizontally into an 80 x 30 grid of 5 km x 5 km cells and vertically into five layers reaching a height of 1100 m above the surface. The domain includes the entire SoCAB, which includes
Orange County and parts of Los Angeles, Ventura, San Bernardino and Riverside Counties. The advection/diffusion equation is solved using the Quintic Splines Taylor Series Expansion approach described by Nguyen and Dabdub (2001). Deposition is based on the dry deposition flux described by Wesely for gas-phase species (Wesely, 1989), and a combination of diffusional deposition and gravitational settling for particles (Griffin et al., 2002a).
The current version of the UCI-CIT model utilizes the previous versions of CACM and MPMPO to describe formation of secondary pollutants, including SOA (Griffin et al., 2005, 2002b, a; Pun et al., 2002). Advanced numerical algorithms are used to solve the non-linear system of highly-coupled differential equations involved in a parallel computational architecture (Dabdub and Seinfeld, 1996, 1994; Nguyen and Dabdub, 2002, 2001).

### 2.2  Emissions and Meteorology

Evaluation of aroCACM/MPMPO 1.0 described subsequently is performed with a three-day simulation using meteorological data representative of the meteorology in the SoCAB (Carreras-Sospedra et al., 2010). It is important to note that in the testing and development presented here, a specific historical event is not being simulated.
As part of this work, the UCI-CIT model was updated to accept emissions based on the 2012 Air Quality Management Plan (AQMP) provided by the South Coast Air Quality Management District (SCAQMD, 2013), which include current (2008) emissions. All plots shown are for the second simulation day with 2008 emissions.

### 2.3 Updated Vapor Pressure Calculations and New SOA Lumping Scheme

In the UCI-CIT model, the most recent version of MPMPO is used to calculate gas-particle con-version of secondary organic species (Griffin et al., 2005, 2003). In species-specific equilibrium-partitioning based models, including MPMPO, thermodynamic properties including sub-cooled liq-uid vapor pressure ($p_L^O$) are required to determine the extent of gas-particle conversion (Pankow, 1994). However, experimental values for $p_L^O$ are often unavailable, and methods for estimating $p_L^O$ based on molecular structure and/or other properties are required. The SIMPOL.1 group-contribution method of Pankow and Asher (2008) is adopted here to calculate updated SOA species vapor pres-sures for use in MPMPO. This method was selected because it includes contributions from functional groups that are present in the SOA species used in MPMPO but were not able to be included using the previously utilized method to calculate $p_L^O$ (Myrdal and Yalkowsky, 1997). In addition, SIM-POL.1 was parameterized and tested using a large set of experimental $p_L^O$ data that spans 14 orders of magnitude (Pankow and Asher, 2008).

Due to computational limitations, individual SOA species of similar molecular structure are of-ten 'lumped' into groups for the purpose of calculating gas-particle conversion in equilibrium-partitioning based models when applied in three-dimensional air quality models. Pun et al. (2002) developed the original SOA lumping scheme used in the UCI-CIT model. Further modifications to the scheme were made by Griffin et al. (2005). Following the inclusion of SIMPOL.1 for $p_L^O$ calcu-lations, the existing SOA lumping scheme used by MPMPO was re-evaluated based on the structure, calculated thermodynamic parameters, and relative abundance of the individual (i.e. non-lumped) SOA species. The motivation behind the specific changes made as part of this work, and their effect on model output are described in Section 3.1.

### 2.4 Aromatic-Derived SOA Formation

The gas-phase mechanism for toluene and *m*-xylene oxidation under varying NO$_X$ conditions origi-nally developed by Xu et al. (2015) was incorporated into the UCI-CIT model without modification. However, gas-particle conversion for the new aromatic-derived SOA species has been adapted for inclusion into a large-scale three-dimensional model. Because of computational limitations in such models, an SOA lumping scheme was applied to the aromatic-derived SOA species, as in the previ-ous version of MPMPO. The new species are divided into five SOA groups labeled C1 – C5. Table 1 describes the new SOA groups and lists the species and surrogates for each group. Structures for these surrogate species are shown in Figure 1. Individual species included in the new SOA groups were selected based on structure and relative yields in the zero-dimensional model simulations of Xu et al. (2015).

As with the pre-existing SOA species, $p_L^O$ for the new surrogate SOA species are calculated using the SIMPOL.1 method described in section 2.3. These new, highly oxygenated SOA groups also

could be expected to partition favorably into the aqueous phase. The MPMPO is a fully coupled mixed-phase SOA model that accounts for partitioning of each SOA group into an aqueous and organic aerosol phase simultaneously (Griffin et al., 2005, 2003). Therefore, Henry's Law constants (HLC) for the new species were calculated using the method of Suzuki et al. (1992) for use in the aqueous-phase partitioning module of MPMPO. Finally, due to the new grouping scheme, the recalculated $p_L^O$, and the introduction of aqueous-phase partitioning for the new SOA species, the experimentally determined correction factors for $p_L^O$ used by Xu et al. (2015) were not adopted in aroCACM/MPMPO. This could result in an underestimation of SOA formation. To explore further this potential underestimation of SOA, several simulations were performed in which correction factors were applied to bring the SIMPOL.1-calculated $p_L^O$ into agreement with those used by Xu et al. (2015) at 298 K. Results from these simulations are discussed in section 3.2.

Emissions of aromatic compounds in the 2012 AQMP are grouped into high-SOA-yield (AROH) and low-SOA-yield (AROL) species. In the absence of more detailed speciation data, some estimation of the contribution of toluene and *m*-xylene to total aromatics must be made. To evaluate the impact of toluene and *m*-xylene oxidation on SOA concentrations individually, model runs are presented in which AROH emissions are assumed to be entirely toluene or entirely *m*-xylene. A scenario where toluene and *m*-xylene are assumed to be 30% each of AROH is also presented, as an approximation of the impact of this new chemistry in a real urban environment. Although this is a rough estimate, toluene and *m*-xylene are known to be major contributors to aromatic emissions in urban areas (de Gouw et al., 2005), and other aromatics, including *o*- and *p*-xylene, likely exhibit similar reactivity. Some aromatic compounds that are present in the atmosphere, such as styrene, are expected to have higher SOA yields due to their reaction with ozone (Hallquist et al., 2009). However, because of their relatively low concentrations, these species are not specifically treated in this work. Unless otherwise noted, all results shown for model simulations that include the new aromatic-derived SOA species use this final emissions scenario that includes *m*-xylene and toluene.

## 2.5 Low-NO$_X$ Chemistry

Ashworth et al. (2015) reported several updates to CACM (what they termed CACM0.0), primarily related to low-NO$_X$ chemistry, based on a comparison to field measurements from a forest canopy and to other gas-phase chemical mechanisms. While specific monoterpene oxidation chemistry for $\alpha$-pinene, $\beta$-pinene, and limonene is not included in the UCI-CIT version of CACM, two of their recommended adjustments were made. The rate constants for the reaction of organic peroxy radicals (RO2) with hydroperoxy radicals (HO2) or other RO2 species were increased to be in agreement with Table A2 in Ashworth et al. (2015). Additionally, the formation of isoprene-derived organic nitrates, along with their subsequent reactions with OH, were added. However, these modifications had little effect on modeled NO and $NO_2$ concentrations or SOA concentration. This would be expected as

biogenics play a much greater role in SOA formation in a forested region (for which their model was
developed), than in the relatively high-$NO_X$, arid SoCAB region.

## 3 Results and Discussion

### 3.1 Updated Vapor Pressure and New SOA Lumping Scheme

The change in predicted SOA upon implementation of the SIMPOL.1 method for vapor pressure
calculations is shown in Figures 2 and 3. Use of the SIMPOL.1 vapor pressure calculations results in
a modest increase in SOA concentration over the base case (Fig. 2a,b). As seen in Figure 3(a,b), this
increase can be attributed primarily to SOA group B4, which includes oxidation products of long-
chain alkanes (Griffin et al., 2005; Pun et al., 2002). Because of the large contribution of SOA group
B4 to total SOA concentration, a redistribution of SOA species was considered and performed. A
summary of modifications to the SOA lumping scheme is presented in Table 2. The structures of the
resulting 12 surrogate SOA species are shown in Figure 4.

The new redistribution results in a large decrease in total SOA concentration (Fig. 2c). The contri-
bution of each SOA group to total SOA in the redistributed scheme is shown in Figure 3c. Splitting
SOA group B4 (long-chain alkane oxidation products) into groups B4, B6 and B7 (Table 2) results
in an order-of-magnitude decrease in SOA from these species. Previously, the surrogate species for
group B4 had both the lowest modeled vapor pressure and lowest domain-wide average concen-
tration of all species in that SOA group. This had resulted in an over-estimation of SOA from this
source, as can be seen by comparing Figures 3b and 3c. The other major effect of the redistribution
of SOA is the decrease in SOA formation from group A2. Although the use of species UR3 as the
surrogate for SOA group A2 results in a lower modeled $p_L^O$ for this group, the largest contribution to
group A2 had come from species UR26, which has a higher $p_L^O$ than the other species in group A2.
Moving UR26 to group A4 and using it as the surrogate for this group results in a decrease in the
combined SOA concentrations from groups A2 and A4.

### 3.2 SOA Formation from Toluene and *m*-xylene Oxidation

Figure 5 shows 24-hr average SOA concentrations assuming AROH emissions are all *m*-xylene, all
toluene, or 30% each *m*-xylene and toluene (with the remaining 40% following the original CACM
AROH chemistry). Comparison with Figure 6c shows that aromatic-derived SOA accounts for 75%
or more of total modeled SOA concentration in the domain. In these simulations, *m*-xylene oxida-
tion leads to higher total SOA concentration than does toluene oxidation (Fig. 5a,b). The opposite
trend is seen in the simulation results of Xu et al. (2015) and the experiments on which the model
was based, both of which were performed under dry conditions at room temperature. Several factors
may contribute to this difference in results. First, vapor pressure correction factors used by Xu et al.
(2015) were not used here. In addition, the partitioning model here allows for aqueous-phase SOA

formation. Finally, it should be noted that the previous experiments and simulations allowed the parent hydrocarbon to react almost to completion. In fact, the reaction of *m*-xylene with OH occurs more rapidly than does that of toluene, such that a larger fraction of *m*-xylene reacts in that simulation, potentially leading to more SOA formation despite having a smaller yield when observed in the laboratory.

As shown in Figure 6 for the *m*-xylene and toluene scenarios, the speciated SOA compositions are in reasonable agreement with the results of Xu et al. (2015) under high-$NO_X$ conditions. In both cases, there are strong contributions from furanones (C1) and peroxy-nitrates (C3), and in the case of toluene, peroxy-bicyclic rings (C5). However, in both the *m*-xylene and toluene cases, the relative SOA contribution from phenols (C2) is smaller than would be expected from the zero-dimensional model simulations of Xu et al. (2015) for high-$NO_X$ conditions and corresponds more closely to low-$NO_X$ conditions. It is important to note that these conditions are total $NO_X$ and do not generally consider the split of $NO_X$ between NO and $NO_2$.

Because the $NO_X$ levels in the SoCAB are relatively high regardless of time of day, modeled domain average NO and $NO_2$ concentrations are shown in Figure 7a. In general, the contribution to total SOA from groups C1, C2 and C4 are correlated with high NO concentrations, and groups C3 and C5 with lower relative NO concentrations (Fig. 7b) and peak OH concentrations (early afternoon; not shown). In the gas-phase aromatic oxidation mechanism, NO competes with $NO_2$ to react with acyl peroxy radical species. When the [NO]/[$NO_2$] ratio is low, the $NO_2$ reaction forms peroxy-nitrates and leads to an increase in group C3. When the [NO]/[$NO_2$] ratio is high, the NO reaction with certain peroxy radical species can form furanones and lead to an increase in group C1. The reaction of $NO_2$ with phenolic radical species to form nitro-phenols leads to an increase in SOA formation from group C2 when $NO_2$ concentrations are highest. Meanwhile, NO competes with the intra-molecular cyclization reaction of certain peroxy radical species, leading to a decrease in peroxy bicyclic ring species (SOA group C5) under high-NO conditions. The diurnal trend of SOA derived from group C4 is more difficult to explain based on the gas-phase mechanism. Some contribution to the decrease seen in groups C1, C2 and C4 may be due to the increased preference for the gas phase of these semi-volatile species during the daytime, as is supported by the average fraction of species C1, C2 and C4 in the particle phase as a function of time of day (Fig. 8).

Although the experimental validation of the mechanism used in the current work was for dry conditions, aqueous-phase partitioning of aromatic-derived SOA species was incorporated into aro-CACM/MPMPO 1.0. To evaluate the importance of aqueous-phase partitioning for the aromatic-derived SOA species in the updated UCI-CIT model, a simulation was performed in which species C1 – C5 were allowed to partition only to the organic phase. Results indicate that > 99% of modeled SOA from species C1 – C5 is in the aqueous-phase. It is possible that the vapor pressure correction factors used by Xu et al. (2015) would increase the fraction of species C1 – C5 in the organic phase. Therefore, a final simulation was performed in which the SIMPOL.1-calculated vapor pres-

sures were adjusted to match those used by Xu et al. (2015) at 298K. (Note: this model run includes aqueous-phase partitioning of species C1 – C5.) This results in little change to modeled SOA, as seen by comparing Figures 5c and 5d, again suggesting that aqueous-phase partitioning may play a dominant role in SOA formation from the aromatic-derived SOA species. (Note: In the final version of the model, the unadjusted SIMPOL.1 vapor pressures are used.)

In addition to aqueous-phase partitioning, results from other experimental and modeling work suggest that aqueous-phase oligomerization may be an important route to SOA formation from aromatics under atmospherically relevant conditions (Im et al., 2014). Future studies using experiments on aromatic oxidation under a variety of relative humidity conditions will be required to assess further the relative importance of aqueous- vs. organic-phase partitioning and the role of oligimerization and to refine the aroCACM/MPMPO 1.0 model.

## 4  Conclusions

This work reports the development of aroCACM/MPMPO, a gas-phase and SOA model designed for use in large-scale chemical transport models that includes important routes to SOA formation from the oxidation of aromatic species. aroCACM/MPMPO makes use of updated schemes for vapor pressure estimation and grouping SOA species and includes both aqueous- and organic-phase partitioning for all SOA species using an equilibrium-partitioning module. Results presented here from the implementation of aroCACM/MPMPO 1.0 into the UCI-CIT Airshed model support the importance of aromatic-derived SOA in urban areas. Of particular note is the manner in which the concentrations of individual SOA-partitioning species demonstrate different trends with the time of day. Furanones, phenols, and epoxides are found to increase their contribution to total SOA during times of high $NO_X$ and low OH, while peroxy-nitrate and peroxy-bicyclic ring species show the opposite trend. This indicates that even in areas generally considered "high" $NO_X$ it is important to include the contribution of low $NO_X$ pathways to SOA formation from aromatic oxidation, primarily due to the importance of the NO to $NO_X$ ratio.

## 5  Code Availability

Model code is available by request.

*Acknowledgements.* This work was performed under grants from the National Science Foundation (CHE-1213623 and ATM-0901580). Any opinions, findings, and conclusions or recommendations expressed in this material are those of the authors and do not necessarily reflect the views of the National Science Foundation.

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

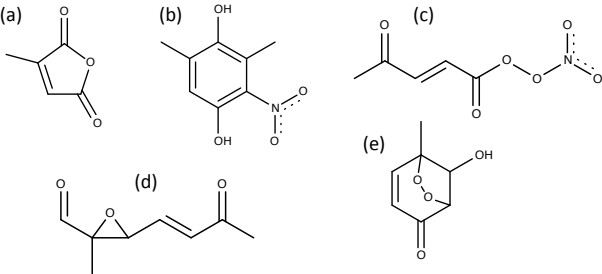

**Figure 1.** Surrogate species for the new aromatic-derived SOA groups: (a) C1, (b) C2, (c) C3, (d) C4 and (e) C5.

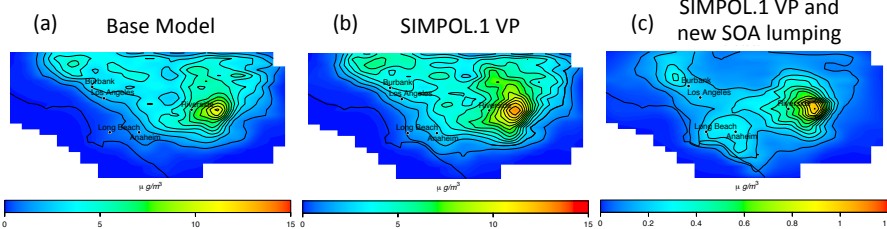

**Figure 2.** Modeled 24-hour average total SOA concentrations (a) before updates to vapor pressure calculations and SOA lumping scheme, (b) after implementation of SIMPOL.1 for vapor pressure calculations, and (c) after adoption of SIMPOL.1 and new SOA lumping scheme. Note that the scale changes between panels.

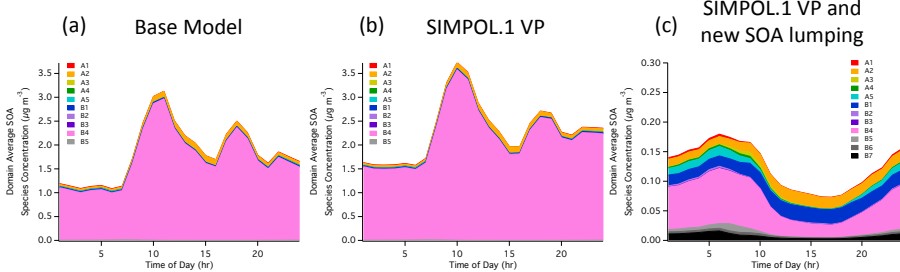

**Figure 3.** Modeled domain-average diurnal SOA concentrations by SOA group (a) before updates to vapor pressure calculations and SOA lumping scheme, (b) after implementation of SIMPOL.1 for vapor pressure calculations, and (c) after adoption of SIMPOL.1 and new SOA lumping scheme. Note that the scale changes between panels. Surrogate structures for each of the lumped SOA species are shown in Figure 4.

**Figure 4.** Surrogate species for each SOA group after changes shown in Table 2, but before addition of new aromatic-derived species: (a) B5, (b) A5, (c) B3, (d) B2, (e) A1, (f) B1, (g) A2, (h) A3, (i) A4, (j) B7, (k) B6, and (l) B4.

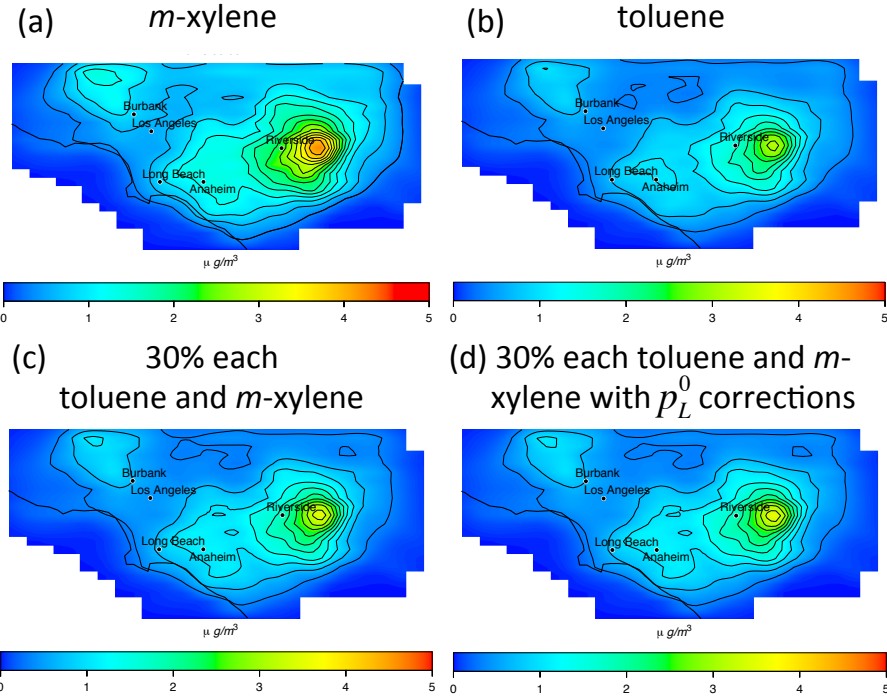

**Figure 5.** Modeled 24-hr average SOA concentrations after inclusion of new aromatic chemistry assuming AROH emissions are (a) all *m*-xylene, (b) all toluene, or (c,d) 30% each *m*-xylene and toluene. Vapor pressures are calculated using (a,b,c) SIMPOL.1, or (d) SIMPOL.1 with SOA species C1 – C5 adjusted to match those used by Xu et al. (2015) at 298 K.

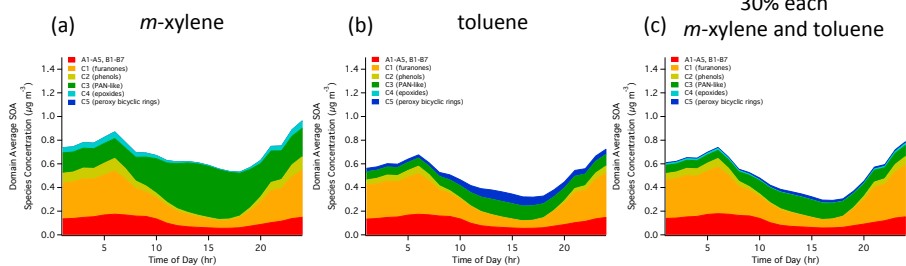

**Figure 6.** Modeled domain-average dirunal SOA concentrations by SOA group after inclusion of new aromatic chemistry assuming AROH emissions are (a) all *m*-xylene, (b) all toluene, or (c) 30% each *m*-xylene and toluene.

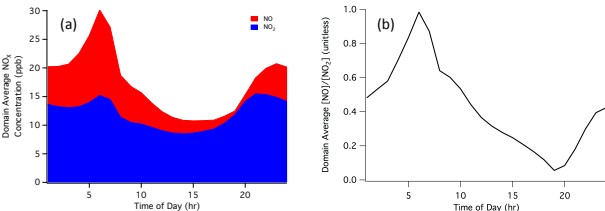

**Figure 7.** Modeled domain-average diurnal (a) NO and $NO_2$ concentrations and (b) [NO] to [$NO_2$] ratio, assuming AROH emissions are 30% each *m*-xylene and toluene.

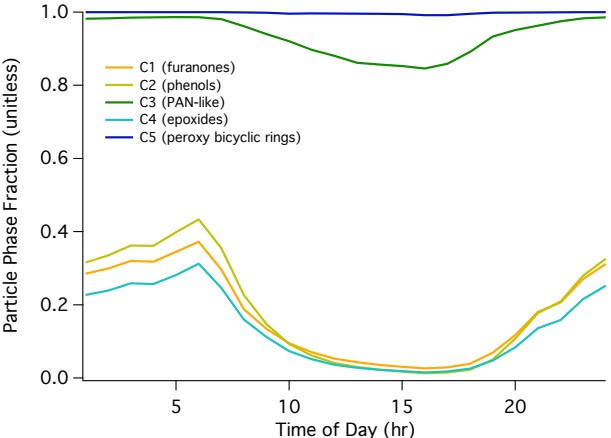

**Figure 8.** Diurnal fraction of aromatic-derived SOA species in the particle phase, averaged over the model domain. This fraction is defined as the mass concentration of a lumped SOA species relative to its total mass concentration (gas- and particle-phase).

**Table 1.** New SOA groups from addition of toluene and *m*-xylene oxidation mechanisms.

| SOA Group | Description | Species Included[1] |
|---|---|---|
| C1 | furanones | UR24, UR72 and UR75[2] |
| C2 | nitro-phenols | RPR4, RP98, UR22, UR57, UR58, UR65, UR66 and UR77[2] |
| C3 | peroxy-nitrates | PN11[2], PN12, PN13 and PN14 |
| C4 | epoxides | R102[2] and UR76 |
| C5 | peroxy-bicyclic rings | RP29, RP30[2] and RP31 |

[1]See Xu et al. (2015) for descriptions of individual SOA species.

[2]Surrogate species

**Table 2.** Updates to the SOA lumping scheme as part of this work, along with their motivation. The original SOA lumping scheme is described in Pun et al. (2002) and Griffin et al. (2005).

| SOA Group | Change | Reason |
|---|---|---|
| A1 | made UR21 (ketopropanoic acid) the surrogate species | Species UR21 has a much higher domain average concentration than the previous surrogate, UR28 (oxalic acid) |
| A2, A4 | moved UR3 from A4 to A2 | The calculated $p_L^O$ for UR3 is closer to species in SOA group A2 |
| A2, A4 | moved UR26 from A2 to A4 | same as above |
| A2 | made UR3 the surrogate species | Species UR3 has representative $p_L^O$ , Henry's law constant, and structure for group A2 and has the highest domain average concentration |
| A4 | made UR26 the surrogate species | same as above |
| B4 | remove UR34 from group B4 | Species UR34 is no longer produced in the gas-phase mechanism (Griffin et al., 2005). |
| B4, B6[1], B7[1] | split group B4 into B4 (AP12), B6 (AP11), and B7 (UR20) | These three SOA species have high domain average concentrations and calculated $p_L^O$ (298 K) that vary by an order of magnitude or more. |

[1]New lumped SOA species