# Peer review of "Development of aroCACM/MPMPO 1.0: A Model to Simulate Secondary Organic Aerosol from Aromatic Precursors in Regional Models"

_Geoscientific Model Development, 2015_

## Referee Comment (RC1) · Anonymous Referee #1 · 2 Feb 2016

**General comments:**

Xu et al. (2015) have further developed the module called aroCACM/MPMPO 1.0 in order to simulate secondary organic aerosol mass (SOA) from aromatic precursors. This study is a follow-up investigation on modifications performed on the module with a special focus on different environmental $NO_x$ conditions and its implementation into the regional air quality model CACM.

The authors focussed on a smaller subset of aromatic species and related gas-phase chemical reactions because of the limited amount of simulation time for any regional

model. While being constrained by simulation capacities earlier studies (Seinfeld and Pandis, 2006; Ng et al., 2007; Xu et al., 2015; Im et al., 2014) have expressed the need to resolve the effect of $NO_x$ on the details of volatile organic compound (VOC) oxidation and subsequent SOA formation. This was considered by concentrationg on two exemplary aromatic spieces only, i.e. toluene and m-xylene as representatives for the other aromatics and their corresponding lumped chemistry. The precursor VOCs and their oxidation products were splitted into low and high-SOA-yield species according to the MPMPO scheme (see Xu et al., 2015). Several improvements recommended by Ashworth et al. (2015) were made. Those include increasing the contribution of hydroperoxy and organic peroxy radical reaction rates by increasing the related reaction rate constants. The new implementation of SIMPOL.1 for estimating compounds saturation vapour pressures was well done and important. And the daily variation of SOA types contributions is quite nice.

**Detailed comments and questions**

Overall, the study by Xu et al. is easy to read and provides some new findings, but is written in a sometimes rapid style and some aspects questions should be clarified in a better way before final acceptation. Those are:

p. 2: Aromatic species were named to react with OH, $NO_3$ and chlorine radicals only. This is true for all the explicitly named and most of the aromatic species except styrene, which may react with ozone too (Atkinson et al., 2006). Although being a minor contribution to the total petrol vapour concentration usually observed the SOA yield by styrene + ozone will be substantially larger as ozone reactions yield substantially less volatile species (Hallquist et al., 2009).

p.5 (general): I would recommend providing a supporting only information document as several essential details of former studies are not directly available. This would help the reader understanding the study details much easier than to read in all the precessing articles. It includes a table of all the SOA groups and corresponding marker species (before and) after the modifications. An overview plot within the study may ease the understanding of the change when splitting B4. Parts of that Table can be found in Tables 1 and 2 as well as in Figure 1. A second aspect worth mentioning covers the initial conditions of the experiments displayed in Figs. 2, 3 and 5-8. Was an initial amount of seed aerosol introduced for partitioning calculations and if so which amount? Or was the system treated clean and the SOA had to form out of the gas-phase via new particle formation? This may cause a changing saturation vapour pressure because of Kelvin effects. The consequence would be time delays in experimental observations and challenges for simulation as partitioning requires an at least infinitesimal amount of pre-existing SOA mass. Was there a spin-up time assumed? Did the authors exclude certain data that were out of a certain range? In which concentration range of organic aerosol (OA) the authors would classify the approach to work properly and in which not?

p. 8: I would favour some "rapid writing style" improvements such as for "Ashworth et al. (2015) reported several updates to CACM (what they termed CACM0.0)...The rate constants for the reaction of organic peroxy radicals ($RO_2$) with hydroperoxy radicals ($HO_2$) or other $RO_2$ species were increased to be in agreement with Table A2 in Ashworth et al. (2015).". Both rate constants were increased by which factor etc? Was the change substantial i.e. affecting any of the results displayed later on? Note Ashworth et al. (2015) did a forest study while this study deals with urban areas and does not include other biogenic VOCs than isoprene.

p.9 (general): The redistribution of class B4 to subspecies and the related results left me somewhat puzzled when looking at Figure 3. While the saturation vapour pressure estimation had a notable (+1 $\mu g/m^3$) effect the daily pattern remained unchainged.

[Figure]

Both rush-hours are visible at different intensity, which agrees with the different mixing layer height and dilution. This daily pattern is not obtained for the simulation of the redistributed B4 SOA class that basically shows an inverse daily structure of a mixing layer height and substantially lower formation yields. In typically elevated urban conditions I would expect to obtain similar results for a lumped compound class and for the individual compounds if using the Pankow (199a, 1994b) approach, since there is no condensation but partitioning taking place at any concentration level. My two only explanations for that would be (i) one of both approaches used a completely different saturation vapour pressure or (ii) the preexisting organic aerosol mass is subestimated with a negative feedback on the formation rate. Thus, I don't know which of both simulations to trust as no observations are displayed for quality check. For a better understanding I would recommend a replacement of the class numbers like A1 by a structure based name in Fig. 3, as the abbreviations have not been explained all in the text and one could easier identify the respondible group for the deviations.

p.11: It makes me struggle somewhat that the authors used nicely the SIMPOL.1 approach (Pankow and Asher, 2008) for a better saturation vapour pressure estimation but adjust it afterwards to match the former results by Xu et al. (2015) at a temperature of 25 degrees C. Did the authors not trust the SIMPOL.1 estimates? Was the approach used to get the relative list (volatile, semi-volatile, non-volatile compounds) in the correct shape but control the absolute values? Please explain.

General: A key feature for every experiment and simulation approach - not this one exclusively - should be a statement in which range a certain approach provides reasonable estimates (valid range). This would force future users to carefully consider not a "black box" for application and take it into account for interpretation of results. Could the authors provide such to make a future application as appropriate as possible and potentially name issues with a need for further improvement? That could serve as new standard.

Finally one question about other SOA precursor and compounds was left: So far larger

biogenic VOCs such as monoterpenes are not included probably because of simulation capacities and lack of informations. Those are much less volatile and would allow a higher preexisting mass present for partitioning of aromatic compounds. Did the authors made tests on the sensitivity of the simulations to preexisting particle mass?

p. 24 and 26 (Fig. 6 and 8): I guess the apparent notable presence of furanones and tiny amounts of epoxides in Fig. 6 can be explained by the primary focus on AROH emissions. Correct?
p. 26 (Fig. 8): The daily structure of PAN-like species and epoxides is very interesting and matches with expectations. Could you provide a standard deviation (variation) of your mean model domain pattern for the 5 classes?

If those issues are clarified the study deserves publication in Geoscientific Model Development.

**References**

Ashworth, K., Chung, S. H., Griffin, R. J., Chen, J., Forkel, R., Bryan, A. M., and Steiner, A. L.: FORest Canopy Atmosphere Transfer (FORCAsT) 1.0: a 1-D model of biosphere– 10 atmosphere chemical exchange, Geosci. Model Dev., 8, 3765–3784, doi:10.5194/gmd-8- 3765-2015, 2015.
Atkinson, R., Baulch, D. L., Cox, R. A., Crowley, J. N., Hampson, R. F., Hynes, R. G., Jenkin, M. E., Rossi, M. J., Troe, J., and IUPAC Subcommittee: Evaluated kinetic and photochemical data for atmospheric chemistry: Volume II – gas phase reactions of organic species, Atmos. Chem. Phys., 6, 3625–4055, doi:10.5194/acp-6-3625-2006, 2006
. Hallquist, M., Wenger, J. C., Baltensperger, U., Rudich, Y., Simpson, D., Claeys, M., Dommen, J., Donahue, N. M., George, C., Goldstein, A. H., Hamilton, J. F., Herrmann,

[Figure]

H., Hoffmann, T., Iinuma, Y., Jang, M., Jenkin, M. E., Jimenez, J. L., Kiendler-Scharr, A., Maen20 haut, W., McFiggans, G., Mentel, Th. F., Monod, A., Prévôt, A. S. H., Seinfeld, J. H., Surratt, J. D., Szmigielski, R., and Wildt, J.: The formation, properties and impact of secondary organic aerosol: current and emerging issues, Atmos. Chem. Phys., 9, 5155–5236, doi:10.5194/acp-9-5155-2009, 2009.

Im, Y., Jang, M., and Beardsley, R. L.: Simulation of aromatic SOA formation using the lumping 25 model integrated with explicit gas-phase kinetic mechanisms and aerosol-phase reactions, Atmos. Chem. Phys., 14, 4013–4027, doi:10.5194/acp-14-4013-2014, 2014.

Ng, N. L., Kroll, J. H., Chan, A. W. H., Chhabra, P. S., Flagan, R. C., and Seinfeld, J. H.: Secondary organic aerosol formation from m-xylene, toluene, and benzene, Atmos. Chem. Phys., 7, 3909–3922, doi:10.5194/acp-7-3909-2007, 2007.

Pankow, J. F., : An adsorption model of the gas/particle partitioning of organic compounds in the atmosphere, Atmos. Environ. 28, 185–188, 1994a.

Pankow, J. F.: An absorption model of the gas/aerosol partitioning involved in the formation of secondary organic aerosol, Atmos. Environ., 28, 189–193, 1994b.

Pankow, J. F. and Asher, W. E.: SIMPOL.1: a simple group contribution method for predicting 20 vapor pressures and enthalpies of vaporization of multifunctional organic compounds, Atmos. Chem. Phys., 8, 2773–2796, doi:10.5194/acp-8-2773-2008, 2008. Seinfeld, J. and Pandis, S.: Atmospheric Chemistry and Physics: from Air Pollution to Climate Change, Wiley-Interscience, Hoboken, NJ, 2006.

Xu, J., Griffin, R. J., Liu, Y., Nakao, S., and Cocker III, D. R.: Simulated impact of $NO_x$ on SOA formation from oxidation of toluene and m-xylene, Atmos. Environ., 101, 217–225, 2015.

---

## Referee Comment (RC2) · Anonymous Referee #2 · 20 Feb 2016

General Comments

In this manuscript the authors report results of a modeling study that incorporates a new gas-phase mechanism for aromatic oxidation, a multiphase partitioning model, updated vapor product pressure estimates, and an improved lumping scheme into an airshed model. The model was used to simulate SOA formation in the Southern California Air Basin. It is generally accepted that aromatic hydrocarbons are an important contributor to SOA formation in urban areas, so developing SOA models for these compounds is important.

The improvements made to the model all seem appropriate. Unfortunately, the atmospheric chemistry of aromatic hydrocarbons is still poorly understood, and their role in SOA formation even more so. Information on reaction product yields are incomplete, for both high and low NOx conditions, and oligomer formation, which adds enormous complexity and uncertainty to the particle chemistry, is known to play an important role in SOA formation. Nonetheless, the approach taken here with regards to incorporating various improvements into the model seem to be about as good a one can currently do, and so are justified. It is to be hoped that future field studies can identify and quantify some of the major SOA compounds that are predicted by this model, thus providing information for better model evaluation. The paper is concise and well written, the model improvements seem straightforward and reasonable, and the results of the test simulations provide some useful information on the likely importance of both high and low NOx chemistry in the region. I think the paper can be published in GMD as is.

Specific Comments

None.

Technical Corrections

None.

---

## Author Comment (AC1)

The authors would like to thank both reviewers for their constructive comments. The manuscript has been updated to address the specific concerns raised. A brief response to the individual comments follows.

Reviewer 1

p. 2: Aromatic species were named to react with OH, NO3 and chlorine radicals only. This is true for all the explicitly named and most of the aromatic species except styrene, which may react with ozone too (Atkinson et al., 2006). Although being a minor contribution to the total petrol vapour concentration usually observed the SOA yield by styrene + ozone will be substantially larger as ozone reactions yield substantially less volatile species (Hallquist et al., 2009).

The chamber experiments on which the model is based were only performed with toluene and *m*-xylene, and thus inclusion of ozone reactions with styrene are beyond the scope of this work. However, two sentences were added to the final paragraph of section 2.4 pointing out this limitation with respect to styrene-derived SOA predictions.

p.5 (general): I would recommend providing a supporting only information document as several essential details of former studies are not directly available. This would help the reader understanding the study details much easier than to read in all the precessing articles. It includes a table of all the SOA groups and corresponding marker species (before and) after the modifications. An overview plot within the study may ease the understanding of the change when splitting B4. Parts of that Table can be found in Tables 1 and 2 as well as in Figure 1.

A table listing the SOA groups, their component species and surrogates before and after the modifications made as part of this work has been added as Supplemental Information.

A second aspect worth mentioning covers the initial conditions of the experiments displayed in Figs. 2, 3 and 5-8. Was an initial amount of seed aerosol introduced for partitioning calculations and if so which amount? Or was the system treated clean and the SOA had to form out of the gas-phase via new particle formation? This may cause a changing saturation vapour pressure because of Kelvin effects. The consequence would be time delays in experimental observations and challenges for simulation as partitioning requires an at least infinitesimal amount of pre-existing SOA mass. Was there a spin-up time assumed? Did the authors exclude certain data that were out of a certain range? In which concentration range of organic aerosol (OA) the authors would classify the approach to work properly and in which not?

It is important to note that no experiments were performed as part of this work. The updated model is based on the work of Xu et al. in which they simulated previously performed (and published) chamber experiments to evaluate a model for SOA formation from aromatic oxidation. One of the objectives of the current work is to

use the Xu et al. model to evaluate the impact of aromatic-derived SOA formation on regional air quality in an urban area. Thus, conditions typical to the South Coast Air Basin of California (SoCAB) were used as initial conditions in the model, which includes some background aerosol. New particle formation from gas-phase precursors typically occurs only when background aerosol concentrations are low, which is not usually the case in urban areas, or under very specific conditions not relevant to the simulated period.

p. 8: I would favour some "rapid writing style" improvements such as for "Ashworth et al. (2015) reported several updates to CACM (what they termed CACM0.0)... The rate constants for the reaction of organic peroxy radicals (RO2) with hydroperoxy radicals (HO2) or other RO2 species were increased to be in agreement with Table A2 in Ashworth et al. (2015).". Both rate constants were increased by which factor etc? Was the change substantial i.e. affecting any of the results displayed later on? Note Ashworth et al. (2015) did a forest study while this study deals with urban areas and does not include other biogenic VOCs than isoprene.

The organic peroxy radicals (RO2) are a class of species and include a number of individual species (RO21, RO22, RO23, etc.). Therefore, their cross reactions and reaction with HO2 involve a large number of reactions, each with rate constants that have been updated according to Ashworth et al. (2015). The modifications related to biogenic oxidation chemistry are described in detail in Ashworth et al. (2015). The implementation here was performed with only the few modifications described in the manuscript. Readers interested in the specific details of their model updates are now referred to their original published work, including the supplemental information table SA2.

Inclusion of the modifications described in Ashworth (2015) had a negligible effect on the model results. This was expected because, as the reviewer states, their work was related to modeling the air above a forest canopy, and biogenics play a much less prominent role in the Los Angeles urban atmosphere. The paragraph describing the effects of their suggested modifications has been updated to make this clear.

p.9 (general): The redistribution of class B4 to subspecies and the related results left me somewhat puzzled when looking at Figure 3. While the saturation vapour pressure estimation had a notable (+1  $\mu$ g/m3) effect the daily pattern remained unchanged. Both rush-hours are visible at different intensity, which agrees with the different mixing layer height and dilution. This daily pattern is not obtained for the simulation of the redistributed B4 SOA class that basically shows an inverse daily structure of a mixing layer height and substantially lower formation yields. In typically elevated urban conditions I would expect to obtain similar results for a lumped compound class and for the individual compounds if using the Pankow (199a, 1994b) approach, since there is no condensation but partitioning taking place at any concentration level. My two only explanations for that would be (i) one of both approaches used a completely different saturation vapour pressure or (ii) the preexisting organic aerosol mass is

**subestimated with a negative feedback on the formation rate. Thus, I don't know which of both simulations to trust as no observations are displayed for quality check.**

The authors agree with the reviewer's explanations. Splitting SOA group B4 into three 'groups' with only one species each had the effect of changing the saturation vapor pressure for the two species that were not the surrogate in the original group B4. In the updated model, their partitioning is calculated based on their own structures using the SIMPOL.1 method, instead of on the structure of the original surrogate (AP12). Thus the updated model can be expected to be more accurate, although there are admittedly no observations with which to compare the model predictions. In general, the Pankow approach using lumped species does a good job at predicting SOA partitioning. However, what we believe our results show is that when species with similar structures, but high concentrations relative to other SOA species, are lumped into the same SOA group, this can have dramatic effects on modeled SOA.

One explanation for the change in diurnal profile of SOA concentration is that the long-chain functionalized alkanes included in the original SOA group B4 (AP11, AP12 and UR20) are strongly correlated with vehicle emissions. Splitting SOA group B4 into groups B4, B6 and B7 results in less contribution to SOA from AP11 (B6) and UR20 (B7) due to their higher SIMPOL.1-modeled saturation vapor pressure. Thus, other SOA species, which may form slowly in the atmosphere, or have sources other than vehicle emissions, contribute more strongly to the net SOA concentration profile. It should also be noted that the time profiles shown in Fig.3 are domain averaged. It is likely that this profile may look different for specific locations within the domain (e.g., downtown Los Angeles).

For a better understanding I would recommend a replacement of the class numbers like A1 by a structure based name in Fig. 3, as the abbreviations have not been explained all in the text and one could easier identify the responsible group for the deviations.

Unlike the new aromatic-derived lumped SOA species, the lumped SOA species developed prior to this work do not lend themselves to a simple structure-based classification, particularly after the re-lumping reported here. A detailed structure-based name for each group would 1) primarily represent only the surrogate species, and 2) be too long for easy inclusion in a figure. However, a reference has been added to the caption for Figure 3 referring the reader to Figure 4 for the surrogate structures for each of the lumped SOA species.

p.11: It makes me struggle somewhat that the authors used nicely the SIMPOL.1 approach (Pankow and Asher, 2008) for a better saturation vapour pressure estimation but adjust it afterwards to match the former results by Xu et al. (2015) at a temperature of 25 degrees C. Did the authors not trust the SIMPOL.1 estimates? Was the approach used to get the relative list (volatile, semi-volatile, non-volatile compounds) in the correct shape but control the absolute values? Please explain. This adjustment was performed just as an "extreme-case" scenario to evaluate the relative importance of aqueous- and organic-phase partitioning of the aromatic-derived SOA species. It is not included in the final version of the model. The paragraph describing this test has been updated to make this more clear.

General: A key feature for every experiment and simulation approach - not this one exclusively - should be a statement in which range a certain approach provides reasonable estimates (valid range). This would force future users to carefully consider not a "black box" for application and take it into account for interpretation of results. Could the authors provide such to make a future application as appropriate as possible and potentially name issues with a need for further improvement? That could serve as new standard.

The range of experimental concentrations used for model development are described in Xu et al. (2015).

Finally one question about other SOA precursor and compounds was left: So far larger biogenic VOCs such as monoterpenes are not included probably because of simulation capacities and lack of information. Those are much less volatile and would allow a higher preexisting mass present for partitioning of aromatic compounds. Did the authors made tests on the sensitivity of the simulations to preexisting particle mass?

It should be noted that monoterpenes and isoprene are included in the reaction mechanism. The lack of model sensitivity to the modifications of Ashworth et al (2015) is evidence that biogenics play a smaller role in the SoCAB region compared to a forest canopy; in addition, the modifications are likely to be relevant in low-NOX scenarios not representative of the SoCAB. Finally, background aerosol concentrations are generally high in this region. Thus, the sensitivities described by the reviewer were not explored, as the focus of this work was specifically SOA formation from aromatic oxidation.

p. 24 and 26 (Fig. 6 and 8): I guess the apparent notable presence of furanones and tiny amounts of epoxides in Fig. 6 can be explained by the primary focus on AROH emissions. Correct?

This follows closely the composition of SOA formed in the chamber simulations of Xu et al (2015). In the CIT model, the AROH emissions were used to estimate the emissions of toluene and m-xylene in the SoCAB region. Thirty percent of the original AROH emissions were changed to toluene, another 30% to m-xylene, and the remaining 40% was left as AROH. The AROH species followed the original oxidation pathways in CACM, and would therefore not contribute to the new SOA groups (C1-C5).

p. 26 (Fig. 8): The daily structure of PAN-like species and epoxides is very interesting and matches with expectations. Could you provide a standard deviation (variation) of your mean model domain pattern for the 5 classes?

A table with average concentration and one standard deviation in concentration for the five aromatic-derived SOA species has been added to the Supplementary information. 
[revised manuscript text omitted]